# Population Dynamics, Fecundity and Fatty Acid Composition of *Oithona nana* (Cyclopoida, Copepoda), Fed on Different Diets

**DOI:** 10.3390/ani11051188

**Published:** 2021-04-21

**Authors:** Fawzy I. Magouz, Mohamed A. Essa, Mustafa Matter, Abdallah Tageldein Mansour, Mohamed Alkafafy, Mohamed Ashour

**Affiliations:** 1Department of Animal Production, Faculty of Agriculture, Kafrelsheikh University, Kafrelsheikh 33516, Egypt; fawzymagouz@yahoo.com; 2National Institute of Oceanography and Fisheries (NIOF), Cairo 11516, Egypt; messa51@yahoo.com (M.A.E.); mhsy4@yahoo.com (M.M.); 3Animal and Fish Production Department, College of Agricultural and Food Sciences, King Faisal University, P.O. Box 420, Al-Ahsa 31982, Saudi Arabia; 4Fish and Animal Production Department, Faculty of Agriculture (Saba Basha), Alexandria University, Alexandria 21531, Egypt; 5Department of Biotechnology, College of Science, Taif University, P.O. Box 11099, Taif 21944, Saudi Arabia; m.kafafy@tu.edu.sa

**Keywords:** live feeds, artificial diets, fatty acid profile, *Oithona nana*, larvae rearing

## Abstract

**Simple Summary:**

Marine larval production is the most critical stage in the life of the marine aquacultured species, which depends on the use of different zooplanktonic organisms as live feed. Copepods are high-quality live prey that could be efficiently used to overcome the transition period from live food to weaning with an artificial diet in the post-larval stages. The main culture systems of copepods use microalgae as uni-food, nevertheless for the more sustainable and cost-efficient production of copepods, the development of artificial diets is the core of its production techniques. The present study was conducted to improve the production and nutritional quality of copepod, *Oithona nana*, using different diets (soybean, yeast, rice bran, and corn starch). Among all diets, corn starch revealed the highest population growth. Meanwhile, animals nourished by rice bran showed the highest percent of copepodite, nauplii, and fecundity. The nutritional quality of copepods referred to fatty acids profile showed a high percentage of unsaturated fatty acids in copepods fed on rice bran. In conclusion, the dry feeds are very applicable, more economic, and simply alternative diets to substitute microalgae and maximize the fecundity and population of *O. nana* in fish hatcheries.

**Abstract:**

The marine copepod species *Oithona nana* is considered as one of the most successfully mass cultured Cyclopoida species in marine hatcheries. This study investigated the effects of four feed diets (soybean, yeast, rice bran, and corn starch) on the population growth, growth rate, population composition, fecundity, and fatty acid composition of native isolated Cyclopoida copepod species *O. nana*. The experiment was continued for 15 days and the copepods were fed on one of the four diets with a concentration of 1 g 10^−6^ individual day^−1^. The results revealed that corn starch was found to be the most supportive diet for population growth and population growth rate. For nutritional value, copepods fed on rice bran were detected to have the highest content of MUFA, PUFA, and the lowest SFA and SFA/UFA ratio; more importantly, the rice bran diet was the only treatment that showed C20:5ω3. Moreover, copepods fed on rice bran showed the highest significant female fecundity, copepodite, and nauplii percent. Finally, the protocols described in the current study concluded that the dry feeds, especially corn starch, are very useful and applicable in hatcheries for maximizing the fecundity and density of Cyclopoida copepod species, *O. nana*.

## 1. Introduction

Live feeds are the most important basic diet in marine hatcheries [1,2,3]. Many live feed species meet the nutritional requirements of marine larvae, therefore, live feeds are considered a mobile carrier of nutritive substances for marine larvae and postlarvae [4,5,6,7]. 

Copepods are one of the most nutritious live feeds used in marine hatcheries. Despite rotifer and artemia which were extensively used as preys in marine hatcheries [8,9], copepods species are considered the best live prey, due to their higher nutritional value [4,10,11,12,13,14]. The ocean is the major source of cultured copepods and different culture methods of copepods have been referenced in recent years from the World Copepod Culture Database [4]. In truth, there are more than 60 copepod species that have been successfully cultured under laboratory conditions [15], while almost 30 culture methods have been reported [16,17].

In the aquatic environment, copepods serve as trophic linkages in marine ecosystems, connecting primary producers and secondary consumers [10,18,19]. Copepods play important roles in pelagic marine food webs [20], especially gelatinous zooplankton “jellyfish” that usually fed on copepods, which nowadays increase global concern for other important environmental topics, such as plastic pollution, which affect all zooplanktonic organisms [21]. Among copepods, the order Cyclopoida is considered the main organic matter consumer and energy transporter to higher trophic levels, including small fish, larvae, and juveniles of aquatic species in the marine ecosystem. Thus, Cyclopoida represents a major prey source for mesopelagic and bathypelagic fish [19,22]. Cyclopoida copepods, especially species *Oithona nana*, is an excellent source of highly polyunsaturated fatty acids, which make copepods more nutritious and attractive food for larval and small fish [4]. 

Selecting a suitable diet for copepods is a crucial factor that extensively influences the quantity and quality of cultured copepods [23,24,25]. Microalgae are the basic live diet utilized in marine hatcheries for copepod cultures due to many factors, such as nutritional value, size, shape suitability, and digestibility [25,26,27]. In addition, macro and/or microalgal cells are an attractive natural source of bioactive molecules, strongly recommended and needed for the growth and development of copepods and other marine aquatic organisms, including polyunsaturated fatty acid (PUFA), monounsaturated fatty acid (MUFA), saturated fatty acid (SFA), phenols, flavonoids, hydrocarbons, antimicrobial substances [28,29,30]. Lee et al. [31] evaluated the use of different microalgal diets reproductive and productive performance of a Cyclopoida copepod, *Paracyclopina nana.* Five single diets were used including *Phaeodactylum tricornutum* (PHA), *Isochrysis galbana* (ISO), *Tetraselmis suecica* (TET), marine *Chlorella* (MCH), condensed freshwater *Chlorella* (FCH), and the two mixed diets of TET + ISO and TET + PHA, which concluded that females fed TET + PHA and TET + ISO had a higher fecundity than female fed the five single diets. The nauplii fed ISO, TET, TET + PHA, and TET + ISO diets developed into adults, while no nauplii developed into adult when fed MCH, FCH, or PHA. The community of *P. nana* fed PHA was significantly lower than those fed TET, ISO, TET + PHA, and TET + ISO diets. While the community of *P. nana* fed FCH and MCH showed negative growth, however, the high labor and high production cost of microalgae increase the copepods’ price of production. Subsequently, seeking an optimal diet as an alternative to the microalgal diet for copepods species is critical for sustainable fresh and marine aquaculture, in particular for cultured species that have commercial prospects.

Many feeding regimes were referenced as alternatives to the microalgal diet for different cultured copepods and zooplankton species, in general, such as baker’s yeast *Saccharomyces cerevisiae* [32,33], fish diet [34], soybean [27], rice bran [35,36], starch and albumen [37], and glucose [38]. According to the previous references, the alternative feeding regimes for culturing either copepods or zooplankton species resulted in adequate population growth and productivity, depending on the cultured species, culture methods, and experimental conditions. However, studying the effects of feed types on copepod quality and quantity is remains necessary to determine their ideal prospect in marine aquaculture.

Cyclopoida copepods, *O. nana*, cultured under controlled laboratory conditions, were enriched with mixed diets of soybean (1 g 10^−6^ ind. 24 h^−1^) and *Nannochloropsis oceanica* (5 × 10^6^ cells mL^−1^) and were utilized as live feed, resulting in improved growth performance, intestine histology, and the economic viability of European seabass (*Dicentrarchus labrax*) postlarvae [39]. Baker’s yeast, *Saccharomyces cerevisiae*, could be successively used as an algal substitute for rotifer in marine hatcheries [40]. In addition, rice bran starch was successfully used as feed for copepods, daphnia artemia, and moina cultures [35,41,42].

To select the optimal feed for the Cyclopoida copepod, *O. nana*, the current study was conducted to evaluate the population growth, population growth rates, population compositions, fecundity, and fatty acid compositions of *O. nana* fed on commercial-grade soybean, yeast, rice bran, and corn starch diets.

## 2. Materials and Methods

### 2.1. Copepods Stock Culture

The marine copepods were isolated from an earthen pond at El-Max Research Station, Alexandria Branch of National Institute of Oceanography & Fisheries, (NIOF), Egypt. During the copepod collections period in spring 2017, the earthen pond temperature (23 ± 2 °C), salinity (31 ± 1 ppt), and pH (7.37 ± 0.10) were recorded at noon. Copepod samples were collected following the protocol described by Abo-Taleb et al. [6]. Isolated individuals were initially examined using a binocular stereomicroscope (Optika Microscopes, B190/B-290, Ponteranica, Italy). Morphological identification and taxonomic characterization were conducted by the Hydrobiology Lab., Marine Environment Division, NIOF. After morphological classification, the isolated adult copepods were identified as Cyclopoida: *Oithona nana* (Figure 1). Adult individuals of *O. nana* were maintained under controlled laboratory conditions (27 ± 1 °C, 20 ppt, pH 7.7 ± 0.15, and continuous gentle aeration) and enriched with soybean and microalga *Nannochloropsis oceanica* NIOF15/001 (5 × 10^6^ cells mL^−1^).

### 2.2. Regime and Experimental Design

Commercial grade soybean, yeast (*Saccharomyces cervicates*), rice bran, and corn starch were used as feed for the copepods. Yeast (*S. cervicates*) and corn starch were supplied by the Starch and Yeast Company, Egypt, while soybean and rice bran were supplied by the Fish Feed Factory located in Alexandria, Egypt. The copepod individuals were divided into 4 groups with 4 various diets: soybean, yeast, rice bran, and corn starch, with concentration of 1 g for 10^6^ individual for 24 h of each. To prepare the concentration of soybean, yeast, rice bran, and corn starch (three replicates for each treatment), 1 g of a commercial *S. cervicates*, corn starch, a very finely grounded commercial soybean, and rice bran were dissolved separately in 100 mL of warm water (35–45 °C), shaken vigorously, and then blended using a kitchen mixer until fresh instant emulsion was formulated to be used for the enrichment of the different treatments in the feeding regimes [27]. The experiment was continued for 15 days.

The density of copepods in each group was estimated as individual mL^−1^ and the needed concentration for each group was estimated depending on the previously accounted copepods every three days (day-0, day-3, day-6, day-9, day-12, and day-15). Before the start of the experiment, the copepods were harvested from the stock culture tank and transferred to the new culture water for a 24 h gut evacuation to prevent the effects of resident soybean and algal diet [25,43]. At the beginning of the experiment, the individuals of adult copepods *O. nana* (average size: 625 µm) were cultured in glass tanks field with 30 L of 1 µm bag-filtered, chlorine-disinfected of diluted seawater (20 ppt) with initial stock density of approximately 1 individual/mL (about 1000 ind. L^−1^). The culture conditions during the experiment were kept under controlled conditions of salinity 20 ppt, temperature 27 ± 1 °C (using a digital thermometer), and pH 7.7 ± 0.15. The tanks were conducted without water replacement and were supplied with gentile aeration to keep dissolved oxygen (DO) over 4 mg L^−1^ (measured using Oxymeter, China). Ammonia (NH_3_) concentration (measured using digital multi-meter, Italy) was < 0.45 ± 0.05 mg L^−1^ in all treatments, and showed no negative effects of food regimes additions.

### 2.3. Tested Parameters

#### 2.3.1. Population Growth, Growth Rate, Composition, and Fecundity

Every three days, 25 mL of culture water from every replicate of each diet was taken to estimate the population growth of copepods, which estimated the increase in the number of animals (ind. mL^−1^). Every 3 days, about one hundred individuals from each replicate were harvested, using a 38 μm mesh, and fixed with a 4% formalin solution to estimate the percentage of population composition and different developmental stages (nauplii, copepodite, male, and female) under a microscope (Optika Microscopes, B190/B-290, Ponteranica, Italy). Twenty to thirty vigorous carrying-females from every replicate were sorted and placed on a Petri dish to examine the fecundity. The population growth rate (*r*) was calculated according to Yin et al. [44], using the following equation:R = (lnN_t_ − lnN_0_)/t(1)
where N_0_ and N_t_ are the initial and final population densities, and t is the incubation time in days.

#### 2.3.2. Fatty acid Analysis

At the end of the experiment (after day-15), all copepods of each replicate were harvested and preserved at −80 °C for fatty acid analysis. The fatty acid of *O. nana* fed different diets were extracted and fatty acids profiles were estimated as described by El-Shenody et al. [45].

#### 2.3.3. Data Analysis

Statistical analyses were analyzed using SPSS Version 16. The results are presented as the mean ± standard error (*n =* 3). All variables were evaluated in three replicates using one-way analysis of variance (ANOVA) followed by Duncan’s multiple range tests to compare the differences among individual means at a significance level of *p* ≤ 0.05.

## 3. Results

### 3.1. Population Growth, Growth Rate, Composition and Fecundity

The population growth (ind. L^−1^), growth rate (*r*), population composition, and fecundity of *O. nana* were significantly affected by different diets (Figure 2, Figure 3, Figure 4 and Figure 5).

Among all experimental diets, corn starch exhibited the highest significant (*p* ≤ 0.05) population growth and growth rate on all investigated days: namely day-3 (2267 Ind. L^−1^ and 0.273, respectively); day-6 (3800 Ind. L^−1^ and 0.445, respectively); day-9 (6267 Ind. L^−1^ and 0.611, respectively); day-12 (7600 Ind. L^−1^ and 0.675, respectively); and day-15 (0.967 Ind. L^−1^ and 0.735, respectively)—followed by yeast and rice bran, while the lowest significant population growth (Figure 2) and growth rate (Figure 3) was observed with soybean.

Figure 4 presented the percentages of the population composition (male, female, copepodite, and nauplii) of *O. nana* fed on different diets. The *O. nana* fed on rice bran diet revealed the highest significant copepodite and nauplii percentages (33.27% and 32.65%, respectively) and the lowest significant male and female percentages (16.50% and 17.58%, respectively), (Figure 4).

Figure 5 showed the fecundity (eggs female^−1^) of *O. nana* fed on different diets. *O. nana* fed on rice bran diet revealed the highest significant fecundity (8.32 ± 0.167 eggs female^−1^), followed by the diet of corn starch (7.67 ± 0.159 eggs female^−1^), while the lowest significant fecundity was found in the diet of soybean (6.17 ± 0.177 eggs female^−1^) and yeast (6.50 ± 0.289 eggs female^−1^).

### 3.2. Fatty Acid Compositions

Fatty acid compositions were significantly varied in *O. nana* fed on the different diets (Table 1). Among all diets, *O. nana* fed on rice bran showed the highest significant (*p* ≤ 0.05) percentages of MUFA (34.87%) and PUFA (1.45%), as well as showed the lowest significant percentage of SFA (63.87%) and the lowest SFA/UFA (1.77), compared to the other diets. Furthermore, *O. nana* fed on rice bran had a greater C18:1c (13.04%), C18:3ω3 (1.20%). No recorded EPA (C20:5ω3) in all *O. nana* fed on the different diets except the diet of rice bran which was the only that revealed a small amount of (0.24%), as presented in Table 1.

## 4. Discussion

Cyclopoida copepods are the most suitable species for mass production [46,47,48]. However, the cost-efficient protocols for the mass production of copepods still need more development [49]. In addition to the low yields, long generation time, seasonal variations of production, and high costs are the main problems limiting the success of the culture of copepods [10,49,50]. Accordingly, there is a need for low-cost dry food to minimize the cost of copepod production [4]. These feedstuffs must be very applicable, more economic, and simply have the potential to be alternative diets to substitute microalgae, and maximize the fecundity and population of Cyclopoida copepods, *O. nana* [39].

In the current study, different diets (soybean, yeast, rice bran, and corn starch) were evaluated to select the suitable diet for maximizing the quantity (population growth, population growth rate, population composition, and fecundity) and quality (fatty acid composition) of *O. nana.* The commercial grades of soybean, yeast, rice bran, and corn starch did not show any negative effects on the water quality of the experimental aquariums. The growth and community composition of *O. nana* in the present study was comparable with the previous studies using macroalgae as a diet [31,51]. Meanwhile, each dry diet has a different pattern of modulating *O. nana* growth and quality. For instance, soybean, which is characterized by high protein percentage, appealing smell, low price [52], and low content of specific anti-nutrients, including oligosaccharides and allergic proteins [53]. However, the growth of *O. nana* in the present study with soybean had the lowest rate compared with other tested diets, which could be due to the high concentration of soybean meal, or nitrogen wastes and induced toxicity [54]. In the study of El-khodary et al. [27] who found that *Cyclops* fed on soybean had high density (27 individuals mL^−1^) but this diet provided low nutritional value compared with those fed on microalgae.

In marine hatcheries, Baker’s yeast, *Saccharomyces cerevisiae*, could be successively used as an algal substitute. Moreover, it has obvious benefits, like the reduction in algal production facilities and subsequently reducing the production cost. Many authors cited that the low concentration of yeast did not influence water quality [27,32,33]. Rice bran was successfully used as feed for copepods, daphnia, artemia, and moina cultures [35,41,42]. Amian et al. [36] cited that the diversity and abundance of Copepoda, Rotifera, and Cladocera were enhanced when using rice bran in fishponds during the rearing of tilapia. Rice bran, as well as soybean, must be processed into small particle suspension to fit the mouth of the cultured copepods [27,36,42].

In addition, the current findings revealed that the diets significantly affected the quality and quantity of *O. nana.* The highest significant maximum population density and population growth rate were observed with copepods *O. nana* fed on corn starch diet. Our results were in agreement with Sulehria et al. [37], who cited that rotifers cultured with corn starch resulted with relatively high growth. As cited by Tester et al. [55], the starch grains are composed of two types of alpha-glucan, amylopectin and amylose, which substitute approximately 98%–99% of the dry weight of starch, moreover, starch contains relatively low quantities (0.4%) of minerals. Moreover, the current study indicated that *O. nana* fed on rice barn diet had a higher fecundity (8.33 eggs female^−1^) compared with those fed on soybean, yeast, and corn starch (6.17, 6.50, and 7.67, respectively). Carli et al. [56] reported that the type of diets is strongly affecting the fecundity and survival of Harpacticoida Copepoda, *Tigriopus fulvus*. Previous studies have confirmed that the fecundity of copepods can probably be linked to the content of PUFA in their diets [23,25,57]. The importance of dietary PUFA contents to the fecundity of *O. nana* was detected in the experimental diets.

In the current study, the fatty acids composition of copepods fed on a rice bran diet may explain the increase in female fecundity obtained by this diet, compared to the other diets. In the current study, comparing to the fatty acid compositions of copepods fed on soybean, yeast, and corn starch, the fatty acid composition of copepods fed on rice bran resulted in the highest significant MUFA and PUFA, as well as the lowest significant SFA and the lowest SFA/UFA ratio. These findings may be due to the nutritional value of rice bran [35,41]. Bhat et al. [58] cited that commercial grade rice bran contains 50% carbohydrate, 15% protein, 20% fatty acids (linoleic and oleic acids), 5% Vitamin E (tocatrienols, tocopherols, oryzanols, phytosterols), and a low amount of other micronutrients.

The fatty acid profiles of copepods and its diets are very important, not only for assessing their aquaculture potential [24,59], but also for understanding and investigating the trophic ecology of copepods [60,61]. Many authors cited that copepods have the ability to endogenously synthesize PUFA from short-chain fatty acids [31,62]. The ability of fatty acid transformation in copepods is species-specific [51]. Harpacticoida and Cyclopoida copepods are reported to be able to synthesize the long-chain fatty acids, especially EPA and DHA, from dietary short-chain fatty acids [51,62,63]. Interestingly, the copepods fed on rice bran are the only exhibited EPA in their fatty acid profile compared to the copepods fed on other diets. These findings may be attributed to the nutritional value of rice bran. Moreover, the EPA contents of copepod fed on rice bran may explain the high obtained fecundity, as well as the high significant percentage of nauplii (32.65%) and copepodite (33.27%) in the population composition of *O. nana*. In current study, although utilizing dry feeds by *O. nana*, the previous findings may explain the improving in fecundity and density of *O. nana*. The observed trend of fecundity was similar to the examination of population composition, since rice bran provided the highest total percentage of nauplii and copepodite population. In addition, the rice bran diet was markedly predominated by nauplii and copepodite, with relatively few adult males and/or females, as shown in (Figure 4). Our finding is in agreement with the results of Pan et al. [25] who cited that the high PUFA content of the diet positively affected the fecundity, nauplii, and copepodite population of Copepoda, *Apocyclops royi;* moreover, he recommends that the marine microalgae *Isochrysis galbana* and *Nannochloropsis oculata* are the optimal diet for supporting the population growth and superior fatty acid profile of Copepoda, *Apocyclops royi.* Therefore, recently, the continuous screening and isolation of aquatic organisms have important implications in terms of sustaining and developing aquaculture [29,39,64].

## 5. Conclusions

The current work concluded that, among the experimental diets (soybean, yeast, rice bran, and corn starch), rice bran is the best diet for culturing *O. nana*, since it achieved a high percentage of nauplii and copepodite, maximum fecundity, and improved the fatty acid profiles, especially the PUFA content of *O. nana*. On the other hand, the corn starch was more suitable for improving the maximum population density and the maximum growth rate of cultured *O. nana*. Finally, it may be concluded that the dry feeds protocol described in the present study are very useful and applicable for maximizing the fecundity and density of *Oithona nana* (Cyclopoida, Copepoda).

## Figures and Tables

**Figure 1 animals-11-01188-f001:**
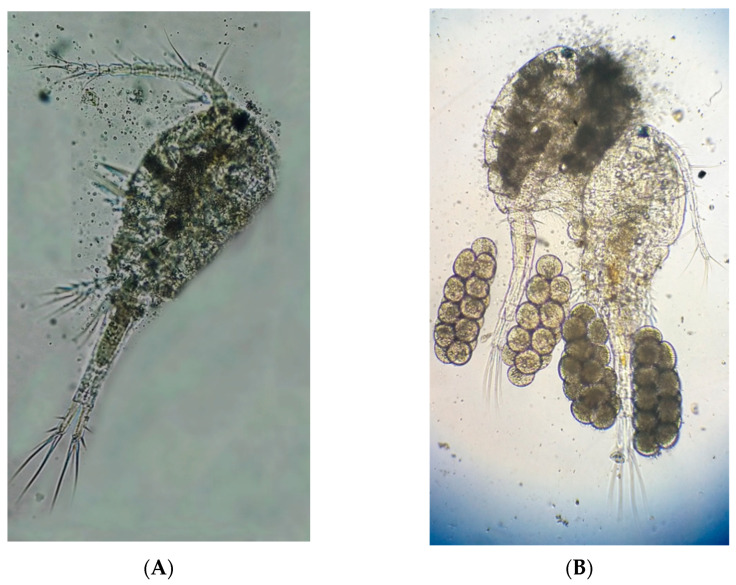
Isolated adult Copepoda, Cyclopoida: *Oithona nana*: (**A**) male; and (**B**) female.

**Figure 2 animals-11-01188-f002:**
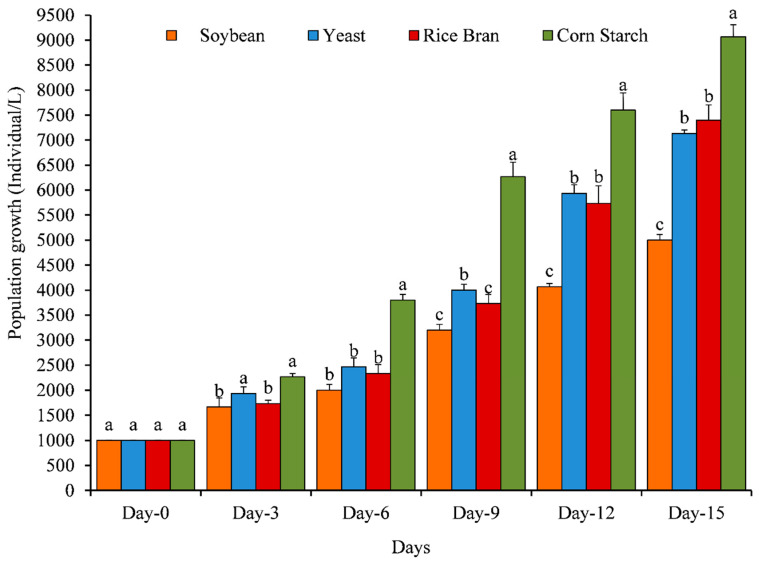
Effect of different diets on the population growth (individual L^−1^) of Cyclopoida copepod, *Oithona nana*. Data are presented as the mean ± standard errors. The letters (a, b, and c) above each bar indicate the significant differences (*p* ≤ 0.05) between different diets at the same day.

**Figure 3 animals-11-01188-f003:**
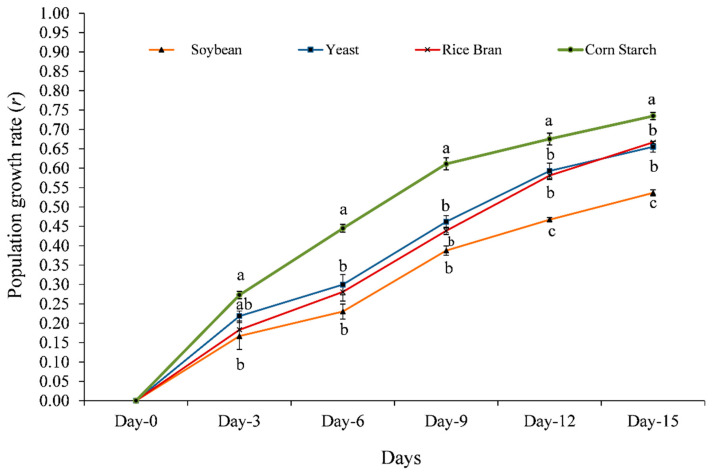
Effect of different diets on the population growth rate (*r*) of Cyclopoida copepods, *Oithona nana*. Data are presented as mean ± standard errors. The letters (a, ab, b, and c) above each bar indicate the significant differences (*p* ≤ 0.05) among different diets on the same day.

**Figure 4 animals-11-01188-f004:**
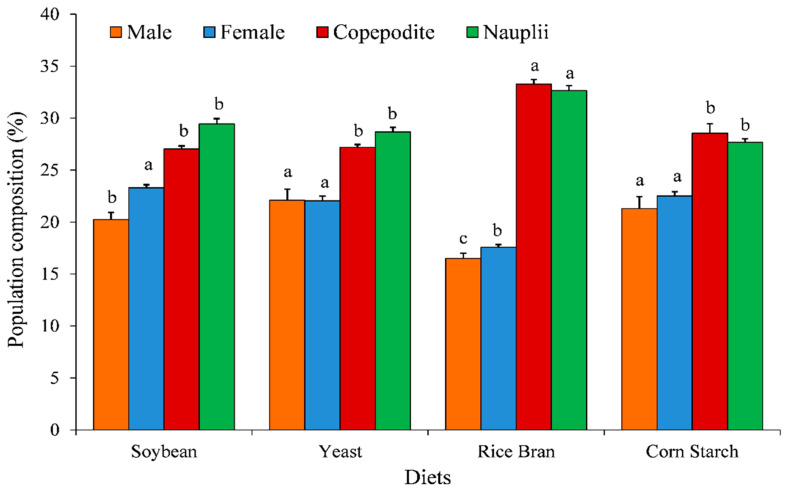
Effect of different diets on mean percentages of population compositions of *Oithona nana*. Data are presented as mean ± standard errors. The letters (a, b, and c) above each bar indicate the significant differences (*p* ≤ 0.05) between the developmental stages (adult males, adult females, nauplii, and copepodites) among the different diets.

**Figure 5 animals-11-01188-f005:**
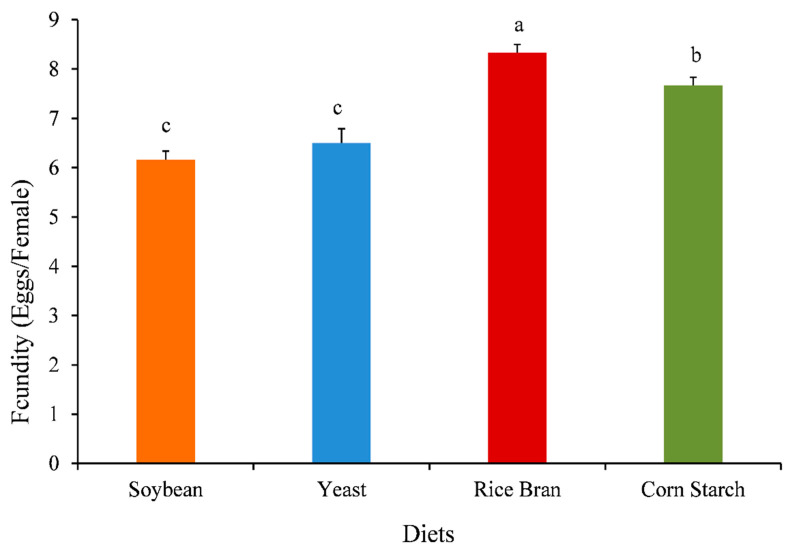
Effect of different diets on fecundity of *Oithona nana*. Data are presented as mean ± standard errors. The letters (a, b, and c) above each bar indicate the significant differences (*p* ≤ 0.05) among the different diets.

**Table 1 animals-11-01188-t001:** Fatty acid profiles of *O. nana* fed on different food regimes.

FA	Soybean	Yeast	Rice Bran	Starch
SFA				
C10:0	0.31 ± 0.006 ^d^	0.46 ± 0.17 ^a^	0.41 ± 0.006 ^b^	0.373 ± 0.003 ^c^
C11:0	0.40 ± 0.006 ^c^	0.81 ± 0.075 ^a^	0.41 ± 0.023 ^c^	0.56 ± 0.006 ^b^
C12:0	0.78 ± 0.003 ^ab^	0.88 ± 0.058 ^a^	0.70 ± 0.020 ^b^	0.52 ± 0.035 ^c^
C13:0	2.17 ± 0.040 ^b^	2.87 ± 0.055 ^a^	2.39 ± 0.205 ^b^	3.10 ± 0.159 ^a^
C14:0	10.51 ± 0.309 ^c^	17.80 ± 0.820 ^a^	14.15 ± 0.471 ^b^	16.10 ± 0.393 ^a^
C15:0	0.91 ± 0.147 ^d^	2.60 ± 0.101 ^b^	1.99 ± 0.012 ^c^	11.80 ± 0.115 ^a^
C16:0	38.54 ± 0.12 ^a^	24.78 ± 0.303 ^c^	30.48 ± 0.029 ^b^	24.83 ± 0.245 ^c^
C17:0	0.55 ± 0.003 ^d^	1.92 ± 0.043 ^a^	1.16 ± 0.015 ^b^	0.69 ± 0.003 ^c^
C18:0	21.88 ± 0.245 ^a^	11.57 ± 0.205 ^b^	9.76 ± 0.064 ^c^	8.12 ± 0.090 ^d^
C20:4	0.83 ± 0.046 ^b^	2.65 ± 0.191 ^a^	2.43 ± 0.300 ^a^	2.22 ± 0.049 ^a^
MUFA				
C14:1	12.35 ± 0.001 ^b^	15.16 ± 0.823 ^a^	15.58 ± 0.003 ^a^	13.93 ± 0.543 ^ab^
C15:1	0.77 ± 0.015 ^b^	1.32 ± 0.001 ^b^	1.16 ± 0.479 ^b^	4.37 ± 0.150 ^a^
C16:1	1.89 ± 0.020 ^b^	4.19 ± 0.592 ^a^	2.44 ± 0.032 ^b^	2.33 ± 0.040 ^b^
C18:1c	6.22 ± 0.150 ^c^	9.07 ± 0.297 ^b^	13.04 ± 0.069 ^a^	9.02 ± 0.433 ^b^
C18:2c	1.04 ± 0.104 ^c^	2.77 ± 0.015 ^a^	2.46 ± 0.006 ^b^	1.12 ± 0.012 ^c^
PUFA				
C18:3ω3	0.83 ± 0.038 ^b^	1.18 ± 0.017 ^a^	1.20 ± 0.061 ^a^	0.91 ± 0.012 ^b^
C20:5ω3	0.00 ± 0.000 ^b^	0.00 ± 0.000 ^b^	0.24 ± 0.006 ^a^	0.00 ± 0.000 ^b^
∑ SFA	76.89 ± 0.297 ^a^	66.33 ± 0.479 ^c^	63.87 ± 0.647 ^d^	68.32 ± 0.084 ^b^
∑ MUFA	22.27 ± 0.260 ^d^	32.49 ± 0.514 ^b^	34.68 ± 5.95 ^a^	30.77 ± 0.095 ^c^
∑ PUFA	0.83 ± 0.037 ^c^	1.18 ± 0.017 ^b^	1.45 ± 0.055 ^a^	0.91 ± 0.012 ^c^
**SFA/UFA**	3.33 ± 0.058 ^a^	1.97 ± 0.043 ^c^	1.77 ± 0.052 ^d^	2.16 ± 0.009 ^b^

Values are means ± SE. Means (*n =* 3) in the same row with different superscript are significantly different (*p* ≤ 0.05). FA: fatty acids; SFA: saturated fatty acids; MUFA: monounsaturated fatty acids; PUFA: polyunsaturated fatty acids; UFA: unsaturated fatty acids.

## Data Availability

The data that support the findings of this study are available from the authors upon reasonable request.

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
