# Peer review of "Population Dynamics, Fecundity and Fatty Acid Composition of Oithona nana (Cyclopoida, Copepoda), Fed on Different Diets"

_animals, 2021, doi:10.3390/ani11051188_

Round 1

Reviewer 1 Report

line 109 - change rice barn to rice bran;

line 115 - change hot freshwater  to water temperature value;

topic 2.2. improve writing;

rewrite the conclusion, it seems like a discussion extension, these two sentences contain your conclusion (due to the high percentage of nauplii and copepodite, maximum fecundity, and improved fatty acid profiles, especially PUFA content of O. nana, the rice bran is the optimal diet for culturing O. nana. While, regarding to improvement of the quantity of cultured O. nana, the corn starch diet is recommended to produce the maximum population density and the maximum growth rate.) 

Author Response

SUMMARY OF AUTHOR(S) RESPONSE TO REVIEWER’S COMMENTS

Manuscript #: animals-1152241

Title of the Manuscript: Evaluation of the population dynamics, fecundity and fatty acid composition of new native isolate Cyclopoida Copepoda, Oithona nana, fed on different diets

Author(s): Fawzy I. Magouz, Mustafa Matter, Mohamed A. Essa, Abdallah Tageldein Mansour, Mohamed Alkafafy, Mohamed Ashour

Reviewer 1# Comment

Author(s) response

Major Comments:

 -        Text Highlighted in yellow: corrected words as a responses to Reviewer #1, 2 and 3

 -        Text Highlighted in green: Added words as results of revising, improving, proofreading, and rewetting of manuscript.

 -     Line 109 - change rice barn to rice bran

Done (Line:134)

 -     Line 115 - change hot freshwater  to water temperature value

Done “warm water (35-45 °C)”, (Line:140).

 -     Topic 2.2. improve writing

Improved and revised

 -     Rewrite the conclusion, it seems like a discussion extension, these two sentences contain your conclusion (due to the high percentage of nauplii and copepodite, maximum fecundity, and improved fatty acid profiles, especially PUFA content of O. nana, the rice bran is the optimal diet for culturing O. nana. While, regarding to improvement of the quantity of cultured O. nana, the corn starch diet is recommended to produce the maximum population density and the maximum growth rate.) 

The conclusion has been rewritten (Lines: 318-325).

We would like to extend our sincere thanks and appreciation to the reviewers and editorial board. In fact, their comments and guidance added a lot to the research and increased its scientific content. Therefore, the words cannot express their gratitude for their time and effort they put in evaluating this research.

Author Response

SUMMARY OF AUTHOR(S) RESPONSE TO REVIEWER’S COMMENTS

Manuscript #: animals-1152241

Title of the Manuscript: Evaluation of the population dynamics, fecundity and fatty acid composition of new native isolate Cyclopoida Copepoda, Oithona nana, fed on different diets

Author(s): Fawzy I. Magouz, Mustafa Matter, Mohamed A. Essa, Abdallah Tageldein Mansour, Mohamed Alkafafy, Mohamed Ashour

Reviewer 2# Comment

Author(s) response

Major Comments:

 -        Text Highlighted in yellow: corrected words as a responses to Reviewer #1,2 and 3

 -        Text Highlighted in green: Added words as results of revising, improving, proofreading, and rewetting of manuscript.

-     This work is useful as copepods may be a significant nutritional source for many other aquaculture species. However the enrichment of Oithona nana with the raw materials used in this experiment (soybean, yeast, rice bran, corn starch), does not qualify its use for marine hatcheries / species due to the low PUFAs and EPA / DHA output profile which is not suitable for marine finfish but may be suitable for freshwater species.

-     Therefore there must be a rephrasing of the Simple Summary, the discussion and conclusions, stating that the used raw materials (soybean, yeast, rice bran, corn starch) can be very useful for maximizing the fecundity and density of Oithona nana cultures, but cannot replace entirely the use of microalgae in marine hatcheries (as microalgae boost the PUFA, EPA, DHA profile of marine finfish larvae).

-     On the other hand, the protocols described in this manuscript, may be very useful and applicable in hatcheries for freshwater species.

-      The aim of this work is to evaluate the possibility of experimented cheap-dry feeds (soybean, yeast, rice bran, and corn starch) as uni-food to reduce the costs of culture and overcome the production difficulties of marine microalgae.

-      It is well known that the microalgal diet is the best basic diet for either zooplankton (rotifer, artemia, and copepods) or early marine larvae, because of many considerations related to its nutritional value, especially its contents of PUFAs, EPA, and DHA, although the culture of microalgae is more difficult and high cost. Therefore, many researchers searching for other feed staff, like cheap-dry feeds, to overcome these obstacles. On the other hand, the maintenance and keeping of the culture of marine copepods did not require qualified feeds contains high content of PUFAs, EPA, and DHA.

-     In addition, in regard to the authors' response to this valuable comment, the simple summary (Lines: 27-29), the discussion (highlighted lines), and the conclusion (Lines: 323-325) parts have been rewritten and stating that the used dry feeds (soybean, yeast, rice bran, corn starch) can be very useful for maximizing the fecundity and density of O. nana cultures as a uni-food to reduce the costs of culture and the production difficulties of microalgae in fish hatcheries.

-      Based on previous work (Lampitt, R.S., 1978. Limnology and Oceanography 23(6):1228-1231), the copepod O. nana is a carnivore feeding on particles with a size between 2.8-400μm. In this work, it is not very clear how the suspensions created for each raw material have been created and how these suspensions have been homogeneous in terms of particle size and density and to which extend O. nana have been able to ingest and digest the particles of the respective suspensions.

-           The protocol of suspension preparing for each dry feed was described in the methods (Lines: 137-142) as the following:  

 “To prepare the concentration of soybean, yeast, rice bran, and corn starch (three replicates for each treatment), 1 g of a commercial S. cervicates, corn starch, a very finely grounded commercial soybean, and rice bran were dissolved separately in 100 ml of warm water (35-45 °C), shaken vigorously, and then blended using a kitchen mixer until fresh instant emulsion was formulated to be used for the enrichment of the different treatments in the feeding regimes”, as modified after (El-khodary et al. 2020).

-           Through this protocol, these suspensions have been completely homogeneous in terms of particle size, density, and homogeneously, and so, they could be able to ingest and digest by Cyclopoida O. nana.

El-khodary, G. M., Mona, M. M., El-sayed, H. S., & Ghoneim, A. Z. (2020). Phylogenetic identification and assessment of the nutritional value of different diets for a copepod species isolated from Eastern Harbor coastal region. The Egyptian Journal of Aquatic Research, 46(2), 173-180

We would like to extend our sincere thanks and appreciation to the reviewers and editorial board. In fact, their comments and guidance added a lot to the research and increased its scientific content. Therefore, the words cannot express their gratitude for their time and effort they put in evaluating this research.

Reviewer 3 Report

Dear Authors,

see my attached report for all my comments and suggestions on your manuscript.

Best regards

The Reviewer

Author Response

SUMMARY OF AUTHOR(S) RESPONSE TO REVIEWER’S COMMENTS

Manuscript #: animals-1152241

Title of the Manuscript: Evaluation of the population dynamics, fecundity and fatty acid composition of new native isolate Cyclopoida Copepoda, Oithona nana, fed on different diets

Author(s): Fawzy I. Magouz, Mustafa Matter, Mohamed A. Essa, Abdallah Tageldein Mansour, Mohamed Alkafafy, Mohamed Ashour

Reviewer 3# Comment

Author(s) response

Major Comments:

 -        Text Highlighted in yellow: corrected words as a responses to Reviewer #1, 2 and 3

Text Highlighted in green: Added words as results of revising, improving, proofreading, and rewetting of manuscript.

The manuscript of Fawzy I. Magouz and colleagues focusing on a very interesting topic related to aquaculture field and more in general to the zooplankton nutritional quality. The study was conducted in an appropriate way with good quality analysis even if the lack of a control group fed with microalgae during the experimental period which correlate the results obtained contextually with the alternative food sources, reduce the validity of the study in my opinion. It would be correct from this point of view to repeat the experiment in order to give it a wider resonance and a real correlation to the studied context and therefore in order to be able to be useful fully to the other researchers of the field.

Alternatively, my advice to the authors is to relate all the manuscript, particularly discussion and conclusion sections, to the results obtained for this species under microalgal feeding regime by other authors, in order to better evaluate the results obtained in this study with other food sources.

The present manuscript giving more resonance to the possibility to substitute microalgae to reduce costs and production difficulties of copepods, but without a real correlation between all these sources it is difficult for the readers well extract this key results.

This is the major point that I have detected on this manuscript to address before the publication.

 See some other minor points in the specific section comments reported below.

Thank you for this important, valuable, and respectable comment. The authors appreciate the advice of Reviewer # 3 and believe that this comment, in particular, will improve the quality of the current manuscript.

The introduction, discussion, and conclusion parts were re-written and linked with the results obtained by Cyclopoida Copepoda, especially O. nana, under the microalgal feeding regime by other authors, as the following:

The introduction: (Lines: 73-87; 100-107).

Discussion: (Lines: 241-243; 250-259; 294-307; 311-316).

Conclusion: has been completely rewritten, (Lines: 318-325).

Title:

The title is informative and descriptive of the report, but in my opinion some hype are avoidable or unnecessary, and it could be more synthetic, for example: “Evaluation of the Population dynamics, fecundity and fatty acid composition of new native isolate Cyclopoida Copepoda, Oithona nana (Cyclopoida, Copepoda), fed on different diets”.

The title was re-written as the reviewer suggest.

Simple Summary-Abstract and Keywords:

Simple Summary and Abstract sections give a good overview of the study, mainly from an introduction and results point of view. Please take care that “Copepoda” refers to a Subclass. The research question is well exposed but the conclusions are missing, please try to insert a conclusive sentence without exceed the words limit imposed by Animals Journal of 200 words.

Inserted. (Lines: 41-43).

Line 17: larval stage is not a bottleneck, to which problem related to this life cycle stage do you refer? Please rephrase this sentence in a more clearly.

Rephrased (Lines: 16-17).

Line 18: substitute zooplanktons with zooplanktonic organisms.

Substituted (Line: 17).

Line 18: why “Copepoda is” and not Copepods are, if you refer to the role of live preys in aquaculture? Please take care of this mistake in all the others similar cases, Copepoda is a Subclass, you cannot refer to it in a generalized way if you write about behaviour and habits of Copepods as organisms.

All manuscript was revised and all “Copepoda” were changed to “copepods” in respect to the context.

Line 27: referred to not “referred as”.

Corrected (Line: 26).

Line 33: O. nana

Corrected (Line: 34).

Keywords: Copepoda, live feeds, artificial diets, population dynamics, fatty acid profile. The highlighted words are already present in the title. Their repetition even in keywords should be avoided, please try to substitute them.

These keywords were deleted and replaced

Introduction:

Introduction section describes sufficiently what is already known about this topic and the research question is clearly outlined. However, it is a little synthetic in my opinion, especially regarding the environmental roles of copepods in aquatic trophic webs, despite the manuscript is related to the aquaculture importance of copepods. For example, between lines 57-62 the authors have not highlighted the role of marine copepods as zooplanktonic organisms for others planktonic trophic levels, as for example gelatinous zooplankton that usually fed on copepods. This role in marine trophic webs is nowadays strictly connected to other important environmental topics as plastic pollution that involved all zooplanktonic organisms. Look and eventually report this reference for better understanding what I mean:

Marco Albano, Giuseppe Panarello, Davide Di Paola, Giovanna D’angelo, Antonia Granata, Serena Savoca, Gioele Capillo. The mauve stinger Pelagia noctiluca (Cnidaria, Scyphozoa) plastics contamination, the Strait of Messina case. Int. J. Environ. Stud. 2021, 00, 1–6, doi:10.1080/00207233.2021.1893489.

Ok, the ecological role of copepods was supported in the introduction section with recent published work. (Lines: 58-63).    

Moreover, some references to the commercial value of the raw materials tested in the study would be necessary (between us and respect the microalgae) for a better overall assessment of the study.

Added (Lines: 100-107).

Line 48: why “nutritional bags”, really are mobile carrier of nutritive substances, bags let me think to something to unmoving.

“nutritional bags” was changed to “mobile carrier of nutritive substances” (Line: 49).

Line 83: O. nana

Changed (Line: 96).

Materials and Methods:

The morphological identification of the species used in this study is, when supported by adequate literature and knowledge, an identification method largely used for long time for zooplanktonic organisms. However, actually a genetic support to this question would be appreciable, if it’s possible to carry out. Moreover, there is a major problem in the experimental design related to the absolute lack of a control group that did not mentioned by authors in material and method section. At the present form they showed a comparison between four experimental alternative diets to the microalgal diets exposed in introduction section as the usual diet of copepods, but a comparison with a control group fed with microalgae and the four experimental diets was not showed. My advice to the authors is to repeat the experiment considering 5 group, including a control group fed with microalgae.

Instead of, the authors were response to the valuable comment of Reviewer # 3 and re-written the introduction, discussion, and conclusion parts and linking our results with the results obtained by O. nana under the microalgal feeding regime by other authors. Moreover, in our next project we will consider this suggestion.

Figure 1: please divide in a and b the two tables with specific description for each one (es. sex, age, growing conditions..) and, if it is possible, use some better quality images.

This Fig. is just to show the isolated adult Copepoda (Cyclopoida: Oithona nana) in the same experimental growing conditions.

Moreover male and female photo has been included with best quality of our efforts.

Line 151: By? A name et al was missed.

Corrected (Line: 177)

Results:

The use of colour figures could be indicated for better presentation of data obtained in the study, at the present form the data presented in some figures are not totally clear (particularly for Figure 3).

The figures are redesigned

Line 174: I think that a 1 was missed in “0967 ind./l”.

Corrected (line: 190)

Line 175: put in black Figure 4.

The figures are redesigned

Line 189: put in black Figure 5.

The figures are redesigned

3.2 Fatty acid compositions: please write out the abbreviations mentioned for the first time.

Done in the introduction part (Lines: 76-77).  

Line 206: except not expect.

Done (Line: 228).

Discussion and conclusions:

Discussion and conclusion not discussed in my opinion the result from multiple angles and answer to the aims of the study. This is due mainly to the absence of a positive control group fed with microalgae to put in correlation with the experimental diets from all the point of view analyzed in the study. In absence of an experimental control carried out from the authors, it is necessary at least put in correlation with the existent bibliography all these aspects, for purpose some alternative diets to substitute microalgal with other more economic and simply to reach raw material.

Thank you for this important, valuable, and respectable comment. The authors appreciate the advice of Reviewer # 3 and believe that this comment, in particular, will improve the quality of the current manuscript.

The introduction, discussion, and conclusion parts were re-written and linked with the results obtained by Cyclopoida Copepoda, especially O. nana, under the microalgal feeding regime by other authors, as the following:

The introduction: (Lines: 73-87; 100-107).

Discussion: (Lines: 241-243; 250-259; 294-307; 311-316).

Conclusion: has been completely rewritten, (Lines: 318-325).

Line 232: a comma between daphnia and artemia was lost.

(Line: 264).

References:

References are relevant, recent and correctly referenced. However, some other studies are needed to enrich introduction section and discussion as highlighted in the related comments on these sections.

Thank you

Weakness summary:

All major and minor points were corrected, respected, and made, according to the best of our abilities

We would like to extend our sincere thanks and appreciation to the reviewers and editorial board. In fact, their comments and guidance added a lot to the research and increased its scientific content. Therefore, the words cannot express their gratitude for their time and effort they put in evaluating this research.

Round 2

Reviewer 3 Report

Dear Authors,

I am very pleased to see that you have quickly and very thoroughly improved the manuscript, also on the basis of my suggestions, really a great job. Congratulations for the seriousness, I am glad to have helped you to enhance your manuscript.

Best regards

The reviewer